# The future of feedback: Motivating performance improvement through future-focused feedback

**Jackie Gnepp[1]\***, **Joshua Klayman[2]**, **Ian O. Williamson[3]**, **Sema Barlas[4]**

**1** Humanly Possible, Inc., Oak Park, Illinois, United States of America, **2** Booth School of Business, University of Chicago, Chicago, Illinois, United States of America, **3** Wellington School of Business and Government, Victoria University of Wellington, Wellington, New Zealand, **4** Masters of Science in Analytics, University of Chicago, Chicago, Illinois, United States of America

\* Jackie@HumanlyPossible.com

**Data Availability Statement:** All relevant data are within the manuscript and its Supporting Information files (S1 Dataset).

**Funding:** This research received funding from the Melbourne Business School while the first three

## Abstract

Managerial feedback discussions often fail to produce the desired performance improvements. Three studies shed light on why performance feedback fails and how it can be made more effective. In Study 1, managers described recent performance feedback experiences in their work settings. In Studies 2 and 3, pairs of managers role-played a performance review meeting. In all studies, recipients of mixed and negative feedback doubted the accuracy of the feedback and the providers' qualifications to give it. Disagreement regarding past performance was greater following the feedback discussion than before, due to feedback recipients' increased self-protective and self-enhancing attributions. Managers were motivated to improve to the extent they perceived the feedback conversation to be focused on future actions rather than on past performance. Our findings have implications for the theory and practice of performance management.

## Introduction

> Once again, Taylor Devani is hoping to be promoted to Regional Manager. Chris Sinopoli, Taylor's new boss, has arranged a meeting to provide performance feedback, especially regarding ways Taylor must change to succeed in a Regional Manager position. Like Taylor's previous boss, Chris is delighted with Taylor's award-winning sales performance. But Taylor was admonished in last year's performance appraisal about cavalier treatment of customers and intolerant behavior toward employees. Taylor was very resistant to that message then and there have been no noticeable improvements since. What can Chris say to get through to Taylor?

This vignette highlights three points that will be familiar to theorists, researchers, and practitioners of performance feedback. First, the vignette reflects that performance feedback often includes a mix of both positive and negative feedback. Second, it reflects the common

authors were either visiting (JG, JK) or permanent (IOW) faculty there. While working on this research, the first two authors (JG, JK) also worked as owners and employees of management consulting firm Humanly Possible. Humanly Possible provided support in the form of salaries and profit-sharing compensation for authors JG and JK, but did not have any additional role in the study design, data collection and analysis, decision to publish, or preparation of the manuscript. The specific roles of these authors are articulated in the "author contributions" section.

**Competing interests:** The authors have declared that no competing interests exist. Authors JG and JK were owners and part-time employees of management consulting firm Humanly Possible during the conduct of this research and the preparation of the manuscript, but the research was not part of their work for the firm, nor did Humanly Possible pay the authors Cover letter– Future of Feedback for their work on this research or otherwise fund it. Humanly Possible has no ownership of, or interest in, any patents or products arising from this research, and no proprietary claim to data or findings from this research. The affiliation with Humanly Possible does not alter our adherence to PLOS ONE policies on sharing data and materials.

experience that the recipients do not always accept the feedback they get, let alone act on it. Third, it raises the question of what a feedback provider should say (and perhaps not say) in order to enable and motivate the feedback recipient to improve.

The present research focuses on feedback conversations in the context of work and career, but it has implications far beyond those contexts. Giving feedback about performance is one of the key elements of mentorship, coaching, supervision, and parenting. It contributes to conflict resolution in intimate relationships [1] and it is considered one of the most powerful activities in education [2]. In all these instances, the primary goal is to motivate and direct positive behavior change. Thus, a better understanding of where performance feedback conversations go wrong and how they can be made more effective is an important contribution to the psychology of work and to organizational psychology, but also to a broad range of psychological literatures, including education, consulting, counseling, and interpersonal communications.

Across three studies, we provide the first evidence that performance feedback discussions can have counterproductive effects by *increasing* the recipient's self-serving attributions for past performance, thereby *decreasing* agreement between the providers and recipients of feedback. These unintended effects are associated with lower feedback acceptance and with lower motivation to change. Our studies also provide the first empirical evidence that feedback discussions promote intentions to act on the feedback to the extent they are perceived as focusing on future performance, rather than past performance. These findings suggest a new line of investigation for a topic with a long and venerable history.

## Performance feedback in the workplace

Performance feedback can be distinguished from other types of managerial feedback (e.g., "production is up 12% from last quarter") by its focus on the recipients' conduct and accomplishments–doing the right things the right way with the right results. It is nearly universal in the modern workplace. Even the recent trend toward doing away with annual performance reviews has come with a directive for managers to have more frequent, if less formal, performance feedback conversations [3].

Psychologists have known for decades that the effects of performance feedback on performance are highly variable and not always beneficial: A meta-analysis by Kluger and DeNisi found that the modal impact on performance is *none* [4]. Such findings fostered a focus on employee reactions to performance appraisals and the idea that employees would be motivated to change behavior only if they accepted the feedback and believed there was a need to improve [5–7]. Unfortunately, unfavorable feedback is not easily accepted. People have been shown to cope with negative feedback by disputing it, lowering their goals, reducing commitment, misremembering or reinterpreting the feedback to be more positive, and engaging in self-esteem repair, none of which are likely to motivate efforts to do a better job next time [8–16].

We are not recommending that feedback providers avoid negative feedback in favor of positive. Glossing over discrepancies between actual performance and desired standards of performance is not a satisfactory solution: Both goal-setting theory and ample evidence support the idea that people need summary feedback comparing progress to goals in order to adjust their efforts and strategies to reach those standards or goals [17, 18]. The solution we propose is feedback that focuses less on diagnosing past performance and more on designing future performance.

## Diagnosing the past

Managers talk to employees about both the nature and the determinants of their performance, often with the goal of improving that performance. Indeed, feedback theorists have long

argued that managers must diagnose the causes of past performance problems in order to generate insight into what skills people need to improve and how they should change [19]. Understanding root causes is believed to help everyone decide future action.

Yet causality is ambiguous in performance situations. Both feedback providers and feedback recipients make causal attributions for performance that are biased, albeit in different ways. Whereas the correspondence bias leads the feedback provider to over-attribute success and failure alike to qualities of the employee [20–22], this bias is modified by a self-serving bias for the feedback recipient. Specifically, feedback recipients are more inclined to attribute successes to their positive dispositional qualities, and failures to external forces such as bad luck and situational constraints [23–26]. These self-enhancing and self-protective attributions benefit both affect and feelings of self-worth [27, 28].

Organizational scholars have theorized since the 1970's that such attribution differences between leaders and subordinates are a likely source of conflict and miscommunication in performance reviews [12, 29–31]. Despite this solid basis in social psychological theory, little evidence exists regarding the prevalence and significance of attribution misalignment in the context of everyday workplace feedback. In the workplace, where people tend to trust their colleagues, have generally positive supervisor-supervisee relations they wish to maintain, and where feedback often takes place within a longer history of interaction, there may be more agreement about the causes of past events than seen in experimental settings. In Study 1, we explored whether attribution disagreement is indeed prevalent in the workplace by surveying hundreds of managers working in hundreds of different settings in which they gave or received positive or negative feedback. (In this paper, "disagreement" refers to a difference of opinion and is not meant to imply an argument between parties.) If workplace results mirror experimental findings and the organizational theorizing reviewed above, then our survey should reveal that when managers receive negative feedback, they make more externally focused attributions and they view that feedback as lacking credibility.

Can feedback discussions lead the two parties to a consensual understanding of the recipient's past performance, so that its quality can be sustained or improved? One would be hard pressed these days to find a feedback theorist who did not advocate two-way communication in delivering feedback. Shouldn't the two parties expect to converge on the "truth" of the matter through a sharing of perspectives? Gioia and Sims asked managers to make attributions for subordinates' performance both before and after giving feedback [32]. Following the feedback conversation, managers gave more credit for success and less blame for failure. However, Gioia and Simms did not assess whether the recipients of feedback were influenced to think differently about their performance and that, after all, is the point of giving feedback.

Should one expect the recipients of workplace feedback to meet the providers halfway, taking less credit for success and/or more responsibility for failure following the feedback discussion? There are reasons to suspect not. The self-serving tendency in attributions is magnified under conditions of self-threat, that is, when information is conveyed that questions, contradicts, or challenges a person's favorable view of the self [33]. People mentally argue against threatening feedback, rejecting what they find refutable [11, 34]. In Studies 2 and 3, we explored the effects of live feedback discussions on attributions, feedback acceptance, and motivation to improve. We anticipated that feedback recipients would find their self-serving tendencies magnified by hearing feedback that challenged their favorable self-views. We hypothesized that the very act of discussing performance would create or exacerbate differences of opinion about what caused past performance, rather than reduce them. We expected this divergence in attributions to result in recipients rejecting the feedback and questioning the legitimacy of the source, conditions that render feedback ineffective for motivating improvement [7, 14, 35].

## Focusing on the future

Given the psychological obstacles to people's acceptance of negative feedback, how can managers lead their subordinates to want to change their behavior and improve their performance? This question lies at the heart of the challenge posed by feedback discussions intended both to inform people and motivate them, sometimes referred to as "developmental" feedback. Despite its intended focus on learning and improvement [36, 37], developmental feedback may nonetheless explicitly include a diagnostic focus on the past [38], such as "why the subjects thought that they had done so poorly, what aspects of the task they had difficulty with, and what they thought their strong points were" (p. 32). In contrast, we propose that the solution lies in focusing on the future: We suggest that ideas generated by a focus on future possibilities are more effective at motivating change than are ideas generated by diagnosing why things went well or poorly in the past. This hypothesis is based on recent theory and findings regarding prospective thinking and planning.

Much prospection (mentally simulating the future) is pragmatic in that it involves thinking about practical actions one can take and behavioral changes one can make to bring about desirable future outcomes [39]. In the context of mixed or negative performance feedback, such desirable outcomes might include improved performance, better results, and greater rewards. Research comparing forward to backward thinking suggests that people find it easier to come up with practical solutions to problems in the future than to imagine practical ways problems could have been avoided in the past: People are biased toward seeing past events as inevitable, finding it difficult to imagine how things might have turned out differently [40–42]. When thinking about their past failures, people tend to focus on how things beyond their control could have been better (e.g., they might have had fewer competing responsibilities and more resources). In contrast, when thinking about how their performance could be more successful in the future, people focus on features under their control, generating more goal-directed thoughts [43]. Thinking through the steps needed to achieve desired goals makes change in the future feel more feasible [44]. And when success seems feasible, contrasting the past with the future leads people to take more responsibility, initiate actions, engage in effortful striving, and achieve more of their goals, as compared to focusing on past difficulties [45]. For all these reasons, we hypothesize that more prospective, forward looking feedback conversations will motivate intentions toward positive change.

## Overview of studies

We report three studies. The first explored the prevalence and consequences of differing attributional perspectives in the workplace. Managers described actual, recently experienced incidents of work-related feedback and the degree to which they accepted that feedback as legitimate. The second study was designed to examine and question the pervasive view that a two-way feedback discussion leads the parties to a shared explanation of past performance and a shared desire for behavior change. We hypothesized instead that the attributions of feedback providers and recipients diverge as a consequence of reviewing past performance. In that study, businesspeople role-played a performance review meeting based on objective data in a personnel file. The third study is a modified replication of the second, with an added emphasis on the developmental purpose of the feedback. Finally, we used data from Studies 2 and 3 to model the connections among provider-recipient attribution differences, future focus, feedback acceptance, and intentions to change. Our overarching theory posits that in the workplace (and in other domains of life), feedback conversations are most beneficial when they avoid the diagnosis of the past and instead focus directly on implications for future action.

## Study 1

We conducted an international survey of managers who described recent work-based incidents in which they either provided or received feedback, positive or negative. We explored how the judgmental biases documented in attribution research are manifested in everyday feedback conversations and how those biases relate to acceptance of feedback. Given well-established phenomena of attribution (correspondence bias, actor-observer differences, self-serving bias), we expected managers to favor internal attributions for the events that prompted the feedback, except for incidents in which they received negative feedback. We hypothesized that managers who received negative feedback would, furthermore, judge the feedback as less accurate and the feedback providers as less qualified, when compared to managers who received positive feedback or who provided feedback of either valence.

### Method

**Participants.**   Respondents to this survey were 419 middle and upper managers enrolled in Executive MBA classes in Chicago, Barcelona, and Singapore. They represented a mix of American, European, and Asian businesspeople. Females comprised 18% of participants. For procedural reasons (see Results), the responses of 37 participants were excluded from analysis, leaving a sample of 382. This study was approved by the Institutional Review Board at the University of Chicago, which waived the requirement for written consent as was its customary policy for studies judged to be of minimal risk, involving only individual, anonymized survey responses.

**Procedure.**   Managers completed the survey online, using the Cogix ViewsFlash survey platform. When they accessed the survey, they were randomly assigned to one of four conditions. Each participant was instructed to think of one recent work-related incident in which they gave another person positive feedback (provider-positive condition), gave another person negative feedback (provider-negative condition), received positive feedback from another person (recipient-positive condition), or received negative feedback from another person (recipient-negative condition). They were asked to describe briefly the incident and the feedback.

The managers were then asked to complete the statement, "The feedback was __% accurate," and to rate the qualification of the feedback provider on a scale from 0 = unqualified to 10 = completely qualified. Providers were asked, "How qualified were you to give the feedback?" whereas recipients were asked, "The person who gave you the feedback—how qualified was he or she to give the feedback?"

Lastly, the managers were instructed to make causal attributions for the incident. They were told, "Looking back now at the incident, please assign a percentage to each of the following causes, such that they sum to 100%." Two of the causes corresponded to Weiner's internal attribution categories (ability and effort) [28]. The other two causes corresponded to Weiner's external attribution categories (task and luck). The wording of the response choices varied with condition. For example, in the provider-positive condition, the response choices were __% due to abilities he or she possessed, __% due to the amount of effort he or she put in, __% due to the nature of what he or she had to do, __% due to good luck, whereas for the recipient-negative condition, the attribution choices were __% due to abilities you lacked, __% due to the amount of effort you put in, __% due to the nature of what you had to do, __% due to bad luck. (Full text is provided in S1 Text.)

### Results

A review of the incidents and feedback the participants described revealed that 25 managers had violated instructions by writing about incidents that were not work-related (e.g.,

interactions with family members) and 12 had written about incidents inconsistent with their assigned condition (e.g., describing feedback received when assigned to a feedback provider condition). The data from these 37 managers were excluded from further analysis, leaving samples of 96, 92, 91, and 103 in the provider-positive, provider-negative, recipient-positive, and recipient-negative conditions, respectively. We tested the data using ANOVAs with *role* (providing vs. receiving feedback) and *valence* (positive vs. negative feedback) as between-subjects variables.

There were three dependent variables: managers' ratings of feedback accuracy, of provider qualifications, and of internal vs. external causal attributions (ability + effort vs. task + luck). Analyses of the attribution variable used the arcsine transformation commonly recommended for proportions [46]. For all three dependent measures, there were significant main effects of role and valence and a significant interaction between them (see Table 1 and Fig 1).

Providers of feedback reported that the incidents in question were largely caused by the abilities and efforts of the feedback recipients. They reported that their feedback was accurate and that they were well qualified to give it. These findings held for both positive and negative feedback. Recipients of feedback made similar judgments when the feedback was positive: They took personal credit for incidents that turned out well and accepted the positive feedback as true. However, when the feedback was negative, recipients judged the failures as due principally to causes beyond their control, such as task demands and bad luck. They did not accept the negative feedback received, judging it as less accurate ($t(192) = 7.50$, $p < .001$) and judging the feedback provider less qualified to give it $t(192) = 5.25$, $p < .001$. One manager who defended the reasonableness of these findings during a group debrief put it this way: "We are the best there is. If we get negative feedback for something bad that happened, it probably wasn't our fault!"

## Discussion

Study 1 confirms that attributional disagreement is prevalent in the workplace and associated with the rejection of negative feedback. Across a large sample of real, recent, work-related incidents, providers and recipients of feedback formed very different impressions of both the feedback and the incidents that prompted it. Despite the general tendency of people to attribute the causes of performance to internal factors such as ability and effort, managers who received negative feedback placed most of the blame outside themselves. Our survey further confirmed that, across a wide variety of workplace settings, managers who received negative feedback viewed it as lacking credibility, rating the feedback as less accurate and the source as less qualified to provide feedback.

These results are consistent with attribution theory and the fact that feedback providers and recipients have access to different information: Whereas providers have an external perspective on the recipients' observable behavior, feedback recipients have unique access to their

**Table 1. ANOVA results for the three dependent measures in Study 1.**

| | Feedback accuracy | | Provider qualifications | | Internal attributions | |
|---|---|---|---|---|---|---|
| | $F$ | $\eta^2$ | $F$ | $\eta^2$ | $F$ | $\eta^2$ |
| Role | 78.0 | .171 | 41.2 | .098 | 49.4 | .115 |
| Valence | 46.8 | .110 | 22.0 | .055 | 41.4 | .099 |
| Role x Valence | 39.6 | .095 | 21.5 | .054 | 44.9 | .106 |

All $F(1, 378)$, all $p < .001$; effect size measures are partial $\eta^2$. Correlations among dependent measures are shown in S1 Table.

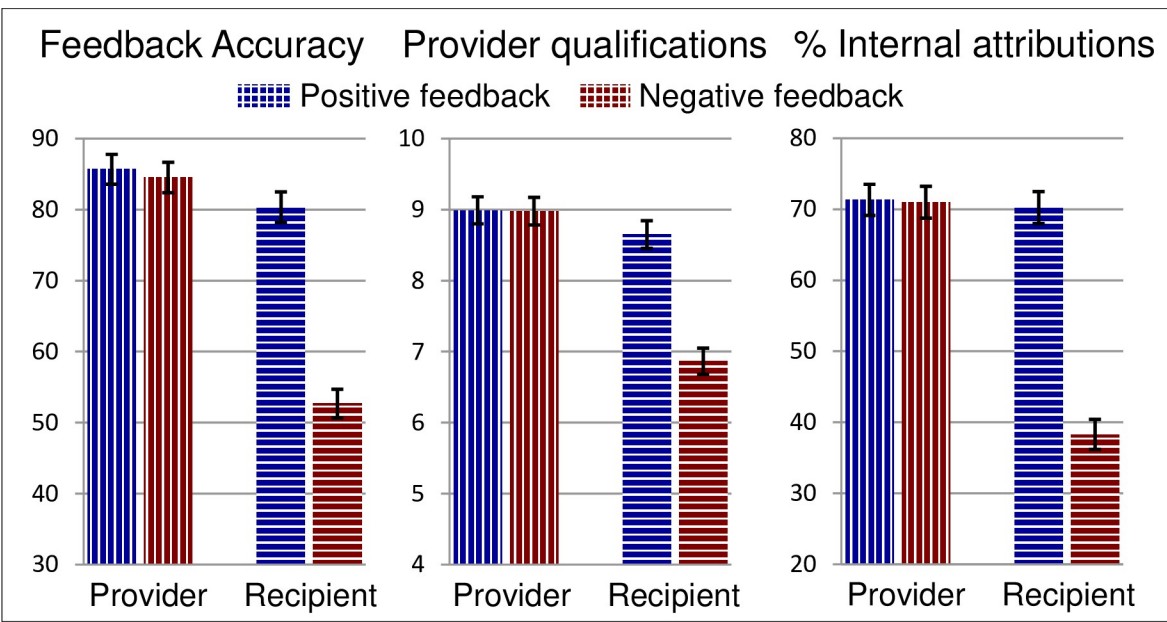

**Fig 1. Ratings of feedback accuracy, provider qualifications, and percent of internal attributions in Study 1.** Results for each dependent variable are shown by role (provider vs. recipient of feedback) and valence (positive vs. negative feedback). Error bars show standard errors.

own thoughts, feelings, and intentions, all of which drove their performance and behavior [24, 47]. For the most part, feedback recipients intend to perform well. When their efforts pay off, they perceive they had personal control over the positive outcome; when their efforts fail, they naturally look for causes outside themselves [48, 49]. For their part, feedback providers are prone to paying insufficient attention to situational constraints, even when motivated to give honest, accurate, unbiased, and objective feedback [20].

In this survey study, every incident was unique: Providers and recipients were not reporting on the same incidents. Thus, the survey method permits an additional mechanism of self-protection, namely, biased selection of congenial information [50]. When faced with a request to recall a recent incident that resulted in receipt of negative feedback, the managers may have tended to retrieve incidents for which they were not to blame and that did not reflect poorly on their abilities. Such biased recall often occurs outside of conscious awareness [51, 52]. For the recipients of feedback, internal attributions for the target incident have direct implications for self-esteem. Thus, they may have tended to recall incidents aligned with their wish to maintain a positive self-view, namely, successes due to ability and effort, and failures due to task demands and bad luck. It is possible, of course, that providers engaged in selective recall as well: They may have enhanced their sense of competence and fairness by retrieving incidents in which they were highly qualified and provided accurate feedback. Biased selection of incidents is not possible in the next two studies which provided all participants with identical workplace-performance information.

## Study 2

In Study 2 we investigated how and how much the feedback conversation itself alters the two parties' judgments of the performance under discussion. This study tests our hypotheses that feedback discussions do not lead to greater agreement about attributions and may well lead to increased disagreement, that attributional misalignment is associated with rejection of

feedback, and that future focus is associated with greater feedback effectiveness, as measured by acceptance of feedback and intention to change. The study used a dyadic role-play simulation of a performance review meeting in which a supervisor (newly hired Regional Manager Chris Sinopoli) gives performance feedback to a subordinate (District Manager Taylor Devani, being considered for promotion). The simulation was adapted from a performance feedback exercise that is widely used in management training. Instructors and researchers who use similar role-play exercises report that participants find them realistic and engaging, and respond as they would to the real thing [32, 53].

The decision to use a role-play method involves trade-offs, especially when compared to studying in vivo workplace performance reviews. We chose this method in order to gain greater experimental control and a cleaner test of our hypotheses. In our study, all participants were given identical information, in the form of a personnel file, ensuring that both the providers and recipients of feedback based their judgements on the same information. This control would not be possible inside an actual company, where the two parties might easily be influenced by differential access to organizational knowledge and different exposure to the events under discussion. Additionally, participants in our study completed questionnaires that assessed their perceptions of the feedback-recipient's performance, the discussion of that performance, and the effects of the feedback discussion. Because this study was a simulation, participants were able to respond honestly to these questionnaires. Participants in an actual workplace performance review might need to balance honesty with concerns for appearances or repercussions; for example, feedback recipients might be hesitant to admit having little intention to change in response to feedback. On the other hand, there are a variety of conditions and motivations that exist in the workplace that cannot be easily simulated in a role-play, such as the pre-existing relationship between the feedback provider and recipient, and the potential long-term consequences of any performance review. Further work will be required to determine how findings from this study apply in workplace settings.

This study comprised two groups that received the same scenarios, but differed with regard to the timing and content of the questionnaires. Recall that the primary goal of Study 2 was to explore how the feedback discussion affects participants' judgments. For this, we analyzed data from the *pre-post* group. Participants in this group completed questionnaires both before and after the discussion. Their post-discussion questionnaire included questions evaluating the conduct and consequences of the feedback discussion, including ratings of future focus and intention to change. A second group of participants (the *post-only* group) completed only a questionnaire after the feedback discussion that did not include future-focus or intention-to-change items. This group allowed us to test whether answering the same questions twice (pre and post the feedback discussion) affected the results.

## Method

**Participants.**   Participants were 380 executives and MBA students enrolled in advanced Human Resources classes in Australia. They represented an international mix of businesspeople: 59% identified their "main cultural identity" as Australian, 20% as a European nationality or ethnicity, 24% Asian, and 12% other; 5% did not indicate any. (Totals sum to more than 100% because participants were able to choose two identities if they wished.) They averaged 35 years of age, ranging from 23 to 66. Females comprised 35% of the sample. Participants worked in pairs. Five pairs were excluded from analysis because one member of the dyad did not complete the required questionnaires, leaving a sample of 117 dyads in the pre-post group and 68 in the post-only group. This study was approved by the Institutional Review Board at the University of Melbourne. Participants' written consent was obtained.

**Materials.** Each participant received a packet of materials consisting of (a) background on a fictional telecommunications company called the DeltaCom Corporation, (b) a description of both their role and their partner's role, (c) task instructions for completing the questionnaires and the role-play itself, (d) a copy of the personnel file for the subordinate, and (e) the questionnaire(s). The names of the role-play characters were pre-tested to be gender neutral. (The full text of the materials is provided in S2–S7 Texts.)

*Personnel file.* The personnel file documented a mixed record including both exemplary and problematic aspects of the District Manger's performance. On the positive side was superior, award-winning sales performance and consistently above-average increases in new customers. On the negative side were consistently below-average ratings of customer satisfaction and a falling percentage of customers retained, along with high turnover of direct reports, some of whom complained of the District Manager's "moody, tyrannical, and obsessive" behavior. Notes from the prior year's performance discussion indicated that the District Manager did not fully accept the developmental feedback received at that time, instead defending a focus on sales success and the bottom line.

*Questionnaires.* Participants in the pre-post group completed a pre-discussion questionnaire immediately following their review of the District Manager's personnel file. They rated the quality of the District Manager's job performance on sales, customer retention, customer satisfaction, and ability to manage and coach employees, using 7-point scales ranging from 1 = Very Low to 7 = Very High. They then rated the importance of these four aspects of the recipient's job performance on 7-point scales ranging from 1 = Not Important to 7 = Very Important. Lastly, participants gave their "opinion about the causes of Taylor Devani's successes by assigning a percentage to each of the following four causes, such that the four causes together sum to 100%." They did the same for "Taylor Devani's failures." Two response categories described internal attributions: "% due to Taylor's abilities and personality" and "% due to the amount of effort and attention Taylor applied." The other two described external attributions: "% due to Taylor's job responsibilities, DeltaCom's expectations, and the resources provided" and "% due to chance and random luck." (We chose the expression "random luck" to imply uncontrollable environmental factors in contrast to a trait or feature of a lucky or unlucky person [54].) Participants chose a percentage from 0 to 100 for each cause, using scales in increments of 5 percentage points. In 4.4% of cases, participants' four attribution ratings summed to a total, $T$, that did not equal 100. In those cases, all the ratings were adjusted by multiplying by ($100 / T$).

Participants in both the pre-post group and the post-only group completed a post-discussion questionnaire following their feedback discussion. This questionnaire asked the participants to rate the favorability of the feedback given, on an 11-point scale from 0 = "Almost all negative" to 10 = "Almost all positive"; the accuracy of the feedback, on a scale from 0% to 100% in increments of 5%; and how qualified the provider was to give the feedback, on an 11-point scale from 0 = "Unqualified" to 10 = "Completely qualified." It continued by asking all of the pre-discussion questionnaire items, allowing us to assess any rating changes that occurred in the pre-post group as a consequence of the intervening feedback discussion. Next, for those in the pre-post group, the questionnaire presented a series of 7-point Likert-scale items concerning the conduct and consequences of the feedback. These included items evaluating future focus and intention to change. Additionally, the post-discussion questionnaires of both groups contained exploratory questions about the behaviors of the individual role-players; these were not analyzed. On the final page, participants provided demographic information about themselves.

**Procedure.** Participants were randomly assigned to dyads and to roles within each dyad. They were sent to private study rooms to complete the procedure. Instructions indicated (a)

15 minutes to review the personnel file, (b) 5 minutes to complete the pre-discussion questionnaire (pre-post group only), (c) 20 minutes to hold the feedback discussion, and (d) 15 minutes to complete the post-discussion questionnaire. Participants were instructed to stay in role during the entire exercise, including completion of the questionnaires. They were told to complete all steps individually without consulting their partner except, of course, for the feedback discussion. The feedback provider was directed by the task instructions to focus on the recipient's "weaknesses as a manager–those aspects of performance Taylor must change to achieve future success if promoted." The reason for this additional instruction was to balance the discussion of successes and failures. Prior pilot testing showed that without this instruction there was a tendency for role-players to avoid discussing shortcomings at all, a finding consistent with research showing that people are reluctant to deliver negative feedback and sometimes distort it to make it more positive [35, 55–57]. When they finished, the participants handed in all the materials and took part in a group debrief of the performance review simulation.

**Design.**   We used analyses of variance to study differences in how the participants interpreted the past performance of the feedback recipient. The dependent variables were participant judgments of (a) internal vs. external attributions for the feedback recipient's performance, (b) the quality of various aspects of job performance, and (c) the importance of those aspects. One set of ANOVAs used post-feedback questionnaire data from both the pre-post and post-only groups to check whether completing a pre-discussion questionnaire affected post-discussion results. The independent variables were *role* (provider or recipient of feedback), *outcomes* (successes or failures of the feedback recipient), and *group* (pre-post or post-only). A second set of ANOVAs used data from the pre-discussion and post-discussion questionnaires of the pre-post group to test our hypothesis that feedback discussions tend to drive providers' and recipients' interpretations of performance further apart rather than closer together. The independent variables in these analyses were *role*, *outcomes*, and *timing* (before or after feedback conversation). In all the ANOVAs, the dyad was treated as a unit (i.e., as though a single participant) because the responses of the two members of a dyad can hardly be considered independent of one another. Accordingly, role, outcomes, group, and timing were all within-dyad variables.

A third set of analyses provided tests of our hypotheses that provider-recipient disagreement about attributions interferes with feedback effectiveness, and that a focus on future behavior, rather than past behavior, improves feedback effectiveness. We conducted regression analyses using data from the pre-post group, whose questionnaires included the set of Likert-scale items concerning the conduct and consequences of the feedback discussion. The dependent variables for these regressions were two measures of feedback effectiveness derived from recipient responses: the recipients' acceptance of the feedback as legitimate and the recipients' expressed intention to change. The predictors represented five characteristics measured from the post-feedback questionnaire: provider-recipient disagreement about attributions, about performance quality, and about performance importance; how favorable the recipient found the feedback to be; and the extent to which the recipient judged the conversation to be future focused.

## Results

**Role differences in the interpretation of past performance before and after feedback discussion.**   Given the results of Study 1 and established phenomena in social psychology, we expected feedback recipients to make internal attributions for their successes and external for their failures more than feedback providers do, to hold more favorable views of their job performance quality than providers do, and to see their successes as more important and/or their failures as less important than providers do. Analyses of the post-discussion ratings in the pre-

**Table 2. ANOVA results for the three measures in Study 2 of interpretations of the feedback recipient's performance.**

| Internal attributions | | | | Performance quality | | | Performance importance | | |
|---|---|---|---|---|---|---|---|---|---|
| | $F$ | $p$ | $\eta^2$ | $F$ | $p$ | $\eta^2$ | $F$ | $p$ | $\eta^2$ |
| Role | 0.27 | .605 | .002 | <u>14.13</u> | <u>< .001</u> | <u>.110</u> | 1.26 | .264 | .011 |
| Outcomes | 1.17 | .281 | .010 | <u>3403.5</u> | <u>< .001</u> | <u>.968</u> | <u>50.34</u> | <u>< .001</u> | <u>.306</u> |
| Timing | 0.03 | .871 | ~ 0 | 3.65 | .059 | .031 | 0.23 | .636 | .002 |
| Role x Outcomes | <u>12.43</u> | <u>.001</u> | <u>.097</u> | 0.89 | .347 | .008 | 4.28 | .041 | .036 |
| Role x Timing | 2.61 | .109 | .022 | 2.16 | .144 | .019 | 3.32 | .071 | .028 |
| Outcomes x Timing | <u>20.97</u> | <u>< .001</u> | <u>.153</u> | <0.01 | .951 | ~ 0 | 1.29 | .258 | .011 |
| Role x Outcomes x Timing | <u>6.46</u> | <u>.012</u> | <u>.053</u> | <0.01 | .967 | ~ 0 | <u>6.43</u> | <u>.013</u> | <u>.053</u> |

$F(1, 116)$ for internal attributions, $F(1, 114)$ for performance quality and importance. Underlined values are effects with $p < .05$ and partial $\eta^2 > .05$.

post and post-only groups (S1 Analyses) confirm those expectations for attributions and for performance quality, but not for performance importance. There were no differences between the pre-post and post-only groups on any of those measures, with all partial $\eta^2 < .02$. Beyond that, we hypothesized that feedback conversations do not reduce provider-recipient differences in interpretation, and may well make them larger. Accordingly, we report here the analyses that include the timing variable, using data from the pre-post group (Table 2).

*Internal vs. external attributions.* Participants in both roles provided attribution ratings before and after the discussion, separately "about the causes of Taylor Devani's successes" and "about the causes of Taylor Devani's failures." There were three significant effects, all of which were interactions. Those were Role x Outcomes, Outcomes x Timing, and Role x Outcomes x Timing. As shown in Fig 2, the three-way interaction reflects the following pattern: The parties

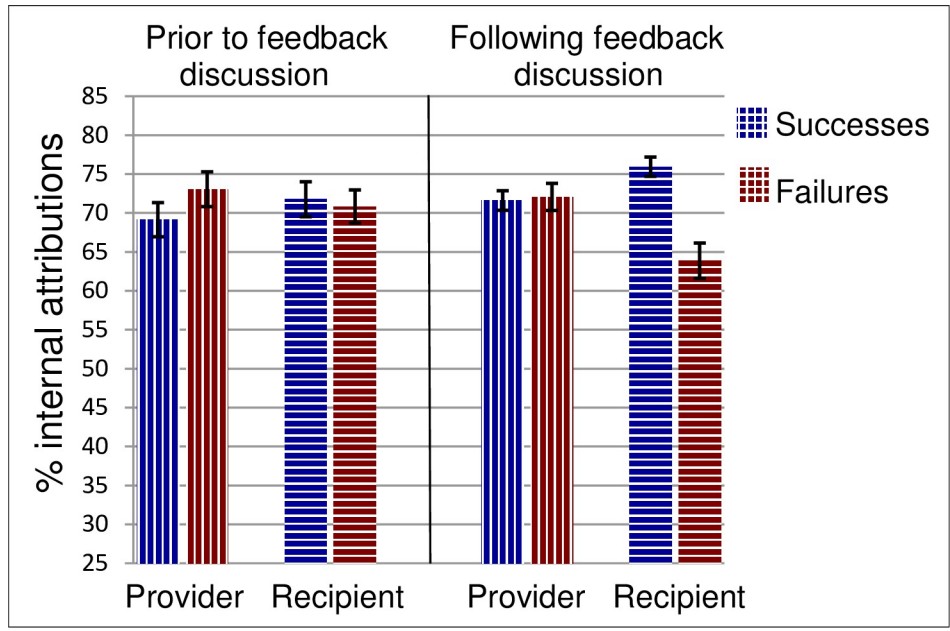

**Fig 2. Percent of performance attributions to internal causes in Study 2.** Results are shown by role (provider vs. recipient of feedback), outcomes (successes vs. failures), and timing (before vs. after feedback). Error bars show standard errors.

began with only minor (and not statistically significant) differences in attributional perspective. Following the feedback discussion however, those differences were much greater. There were no significant effects involving timing for feedback providers: Their attributions changed only slightly from pre- to post-discussion. Feedback recipients, in contrast, showed a highly significant Outcomes x Timing interaction, $F(1, 116) = 19.6$, $p < .001$, $\eta^2 = .14$. Following the feedback conversation, recipients attributed their successes more to internal factors than they did before the conversation and they attributed their failures more to external factors than before ($t(116) = 4.5$, $p < .001$ and $t(116) = 3.3$, $p = .001$, respectively). At the end, the two parties' attributions were well apart on both successes and failures ($t(116) = 2.3$, $p = .024$ and $t(116) = 3.0$, $p = .003$). In sum, the performance review discussion led to greater disagreement between the feedback providers and recipients due to the recipients of feedback making more self-enhancing and self-protecting performance attributions.

*Performance quality.* There were main effects of outcomes and role, but no interactions. As intended, participants rated performance on sales much more highly than they rated the other job aspects (6.72 vs. 3.32 out of 7). Overall, recipients evaluated their performances slightly more positively than the providers did (5.13 vs. 4.91).

*Performance importance.* There was a main effect of outcome, modified by significant Role x Outcomes and Role x Outcomes x Timing interactions. To understand these effects, we followed up with analyses of role and timing for successes and for failures, separately. Feedback recipients rated their successes as more important than feedback providers did (6.41 and 6.12, respectively; $F(1, 115) = 6.20$, $p = .014$, $\eta^2 = .05$), with no significant effects of time. In contrast, importance ratings for failures showed a Role x Timing interaction ($F(1, 114) = 7.77$, $p = .006$, $\eta^2 = .06$): Providers rated failures as more important before discussion, becoming more lenient following discussion (5.75 vs. 5.42; $t(114) = 2.22$, $p = .028$), consistent with the findings of Gioia and Sims [32]. Recipient ratings showed no significant change as a consequence of discussion.

These analyses suggest that in performance conversations, feedback providers do not lead recipients to see things their way: Recipient interpretations of past performance do not become more like provider interpretations. In fact, following discussion, recipients' causal attributions are further from those of the providers. Moreover, across dyads, there was no correlation between the recipient's ratings and the provider's ratings following discussion: Although a ceiling effect limits the potential for correlations on the quality of sales performance (success), the other measures, especially attributions, show considerable variation in responses across dyads but still no provider-recipient correlations (S2 and S3 Tables). For performance quality, performance importance, and attributions, for successes and for failures, all $|r| < .12$ ($p > .22$, $N = 115$ to $117$).

**Effects of attribution disagreement and future focus on recipients' acceptance of feedback and intention to change.**   We hypothesized that provider-recipient disagreement about attributions negatively impacts feedback in two ways, by reducing the extent to which recipients accept the feedback as legitimate, and by reducing the recipient's intentions to change in response to the feedback. We further hypothesized that a focus on future behavior, rather than past behavior, would engender greater acceptance of feedback and greater intention to change. The present study provides evidence for both of those hypotheses.

We measured feedback acceptance by averaging ratings on feedback accuracy and provider qualifications, both scaled 0 to 100 ($r = .448$). We measured intention to change as the average of recipients' responses to three of the Likert questions in the post-feedback-discussion questionnaire ($\alpha = .94$):

Based on the feedback, you are now motivated to change your behavior.

You see the value of acting on Chris's suggestions.

You will likely change your behavior, based on the feedback received.

We analyzed these two measures of feedback effectiveness using regressions with five variables that might predict the outcome of the discussion: post-feedback disagreement about attributions, performance quality, and performance importance (all scored such that positive numbers indicate that the recipient made judgments more favorable to the recipient than did the provider); how favorable the recipient found the feedback to be (rated from 0 = almost all negative to 10 = almost all positive); and the extent to which the recipient thought the conversation was future focused. This last measure is the average of the recipient's ratings on the following three Likert questions on the post-feedback questionnaire (α = .75):

You and Chris spent a large part of this session generating new ideas for your next steps.

The feedback conversation centered on what will make you most successful going forward.

The feedback discussion focused mostly on your future behavior.

We hypothesized that the recipients' acceptance of feedback and intention to change would be affected by the recipients' impressions of how future focused the discussion was. That said, we note that the provider's and the recipient's ratings of future focus were well correlated across dyads ($r(115) = .423$, $p < .001$), suggesting that recipients' ratings of future focus reflected characteristics of the discussion that were perceived by both parties.

As shown in Table 3, recipients' ratings of future focus proved to be the best predictor of their ratings of both feedback acceptance and intention to change. Recipients' favorability ratings also significantly predicted their intention to change and, especially, their acceptance of the feedback. Attribution disagreement between providers and recipients predicted lower acceptance of feedback, but not intention to change. Differences of opinion regarding the quality and importance of various aspects of job performance had no significant effects and, as shown by Model 2 in Table 3, removing them had almost no effect.

## Discussion

As in Study 1, we again observe that the providers and recipients of feedback formed very different impressions about past performance. A new and important finding in this study is that

**Table 3. Effects of future focus, feedback favorability, and provider-recipient disagreements in the interpretation of performance on feedback acceptance and intention to change in Study 2.**

| | Feedback Acceptance | | | | | | Intention to change | | | | | |
|---|---|---|---|---|---|---|---|---|---|---|---|---|
| | Model 1 [.427] | | | Model 2 [.421] | | | Model 1 [.590] | | | Model 2 [.599] | | |
| | Beta | $t(109)$ | $p$ | Beta | $t(113)$ | $p$ | Beta | $t(109)$ | $p$ | Beta | $t(113)$ | $p$ |
| Future focus | .406 | 5.10 | < .001 | .424 | 5.39 | < .001 | .699 | 10.39 | < .001 | .709 | 10.84 | < .001 |
| Favorability | .313 | 3.85 | < .001 | .284 | 3.63 | < .001 | .156 | 2.26 | .025 | .142 | 2.18 | .031 |
| Attribution disagreement | -.207 | -2.68 | .009 | -.173 | -2.39 | .019 | -.014 | -.22 | .828 | -.004 | -.07 | .942 |
| Quality disagreement | -.041 | -.53 | .596 | | | | -.019 | -.29 | .773 | | | |
| Importance disagreement | .102 | 1.39 | .166 | | | | .017 | .27 | .785 | | | |

Model 1 includes all five predictor variables. Model 2 excludes the two that showed no significant effects in Model 1. Numbers in brackets are adjusted $R^2$s.

feedback conversations did not merely fail to diminish provider-recipient disagreements about what led to strong and weak performance; they actually turned minor disagreements into major ones. Recipients made more self-enhancing and self-protective attributions following the performance discussion, believing more strongly than before that their successes were caused by internal factors (their ability, personality, effort, and attention) and their failures were caused by external factors (job responsibilities, employer expectations, resources provided, and bad luck). There were also modest disagreements regarding the quality and importance of different aspects of the recipient's job performance, but these did not worsen following discussion. The most important source of disagreement between providers and recipients then, especially following the feedback conversation, was not about what happened, but about why it happened.

What led recipients of performance feedback to accept it as legitimate and helpful? The best predictor of feedback effectiveness was the extent to which the discussion was perceived as future focused. Unsurprisingly, feedback was also easier to accept when it was more favorable. As predicted, recipients were more likely to accept feedback when they and the feedback providers agreed more about what caused the past events. Greater attribution agreement, however, did not increase recipients' intention to change. These findings suggest that reaching agreement on the causes of past performance is neither likely to happen (because feedback discussions widen causal attribution disagreement) nor is it necessary for fostering change. What does matter is the extent to which the feedback conversation focuses on generating new ideas for future success. We further explore the relations among all these variables following the reporting of Study 3.

## Study 3

Performance feedback serves goals other than improving performance. For example, performance reviews often serve as an opportunity for the feedback provider to justify promotion and compensation decisions. For the recipient, the conversation may provide an opportunity for image management and the chance to influence employment decisions. People may fail to distinguish between evaluation and improvement goals when providing and receiving feedback. In Study 2, the instructions were intended to be explicit in directing participants to the developmental goal of performance improvement, rather than accountability or rewards. Nevertheless, the providers' wish to justify their evaluations and the recipients' wish to influence them might have contributed to the differences we observed in attributions and in judgments about the feedback's legitimacy. To address this concern, we added a page of detailed company guidelines that emphasized the primacy of the performance-improvement goal over the goals of expressing, justifying, or influencing evaluations. There were two versions of these guidelines, which did not differ in their effects.

### Method

Participants were 162 executives and MBA students enrolled in advanced Human Resources classes in Australia. An international mix of businesspeople, 74% said they grew up in Australia or New Zealand, 10% in Europe, 22% in Asia, and 7% other. (Totals sum to more than 100% because some participants indicated more than one.) Participants averaged 39 years of age, ranging from 27 to 60. Females comprised 37% of the participants.

Participants read the same scenario and instructions as in Study 2, with an added page of guidelines for giving developmental feedback (S8 Text). They then completed the same post-discussion questionnaires used for the pre-post group of Study 2, minus the ratings of performance quality and importance for various aspects of the job, which showed no effects in Study

2. (The full text of the questionnaires is provided in S9 and S10 Texts). Taken together, these modifications kept the procedure to about the same length as in Study 2. This study was approved by the Institutional Review Board at the University of Melbourne. Written consent was obtained.

## Results

**Role differences in the interpretation of past performance.** As in Study 2, we calculated the sum of the percentages of attributions assigned to internal causes (ability and personality + effort and attention), applying an arcsine transformation. As before, we analyzed the internal attributions measure with a mixed-model ANOVA treating each dyad as a unit. There were two within-dyads variables: *role* (provider or recipient), and *outcomes* (successes or failures) and one between-dyads variable (guideline *version*). There were no effects involving guideline version (all $F < 1$). The main effects of role ($F(1, 79) = 50.12$, $p < .001$, $\eta^2 = .39$) and outcomes ($F(1, 79) = 113.8$, $p < .001$, $\eta^2 = .59$) and the interaction between them ($F(1, 79) = 86.34$, $p < .001$, $\eta^2 = .52$) are displayed in Fig 3, along with the parallel post-feedback results from the previous two studies. As in Study 2, the two parties' post-discussion attributions were well apart on both successes and, especially, failures ($t(80) = 3.3$ and $9.4$ respectively, both $p \leq .001$). Again, the correlations between the provider's and the recipient's post-conversation performance attributions across dyads were not significant for either successes ($r(79) = -.04$, $p > .69$) or failures ($r(79) = -.13$, $p > .23$) suggesting that conversation does not lead the dyad to a common understanding of what led to good or poor performance.

**Effects of attribution disagreement and future focus on recipients' acceptance of feedback and intention to change.** We conducted regression analyses of the recipient's feedback acceptance and intention to change as in Study 2. The regression models included three predictors: future focus, attribution disagreement, and feedback favorability. Results, shown in Table 4, replicated our Study 2 finding that future focus is the best predictor of both feedback acceptance and intention to change. As before, attribution disagreement predicted lower

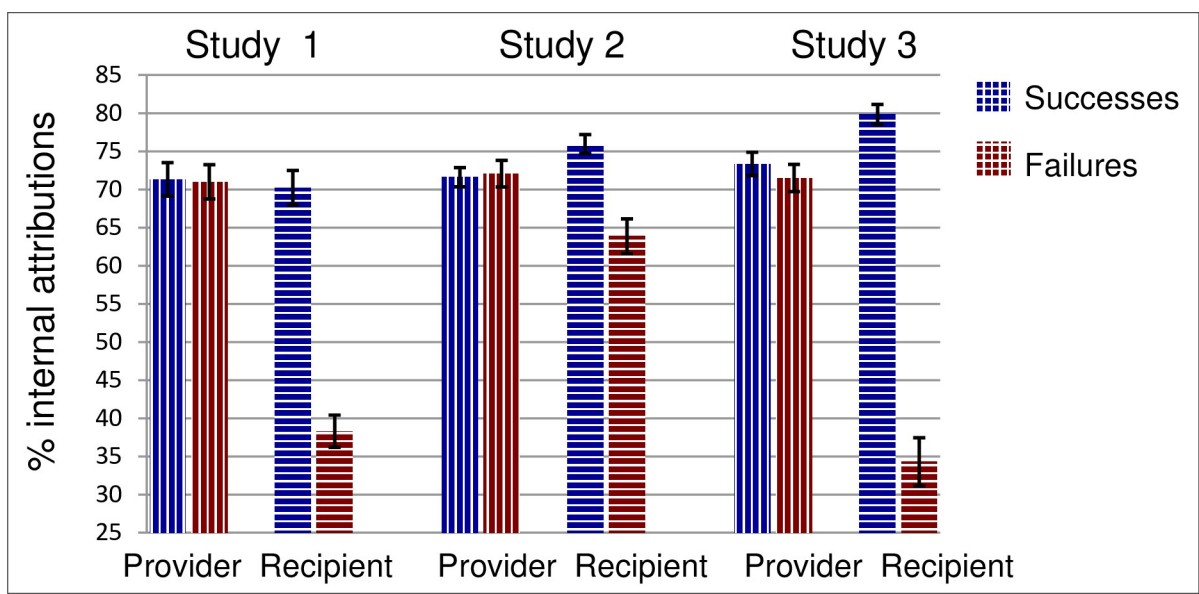

**Fig 3. Percent of performance attributions to internal causes in Studies 1, 2, and 3.** Results are shown by role (provider vs. recipient of feedback) and valence/outcomes (positive feedback for successes vs. negative feedback for failures), following feedback conversation. Error bars show standard errors.

**Table 4. Effects of future focus, attribution disagreement, and feedback favorability on recipients' feedback acceptance and intention to change in Study 3.**

| | Feedback Acceptance [.373] | | | Intention to Change [.323] | | |
|---|---|---|---|---|---|---|
| | **Beta** | *t*(77) | *p* | **Beta** | *t*(77) | *p* |
| Future focus | .411 | 4.432 | < .001 | .549 | 5.697 | .001 |
| Attribution disagreement | -.193 | -2.131 | .036 | -.198 | -2.105 | .039 |
| Favorability | .284 | 3.017 | .003 | -.050 | -.516 | .607 |

Numbers in brackets are adjusted $R^2$s.

acceptance, but in this study it also predicted less intention to change. We again found that feedback favorability ratings were associated with greater acceptance, but this time, not with intention to change. Recipients and providers were again significantly correlated in their judgments of how future focused the conversation was ($r$(79) = .299, $p$ = .007).

## Discussion

Future focus, as perceived by the recipients of feedback, was once again the strongest predictor of their acceptance of the feedback and the strongest predictor of their intention to change. Conversely, attribution disagreement between the provider and recipient of feedback was associated with lower feedback acceptance and weaker intention to change. As in Studies 1 and 2, recipients made more internal attributions for successes than providers did and, especially, more external attributions for failures. The added guidelines in this study emphasizing performance-improvement goals over evaluative ones did not alleviate provider-recipient attribution differences. Indeed, those differences were considerably larger in this study than in the previous one and were more similar to those seen in Study 1 (see Fig 3).

## Future focus, attributions, favorability, and the effectiveness of feedback

The strongest predictor of feedback effectiveness is the recipient's perception that the feedback conversation focused on plans for the future rather than analysis of the past. We seek here to elucidate the relationship between future focus and feedback effectiveness by looking at the interrelations among the three predictors of effectiveness we studied: future focus, attribution disagreement, and feedback favorability.

The analyses that follow include data from all participants who were asked for ratings of future focus, namely those in Study 3 and in the pre-post group of Study 2. We included study as a variable in our analyses; no effects involving the study variable were significant. Nonetheless, because the two studies drew from different samples and used slightly different methods, inferential statistics could be impacted by intraclass correlation within each study. Therefore, we also tested for study-specific differences in parameter estimates using hierarchical linear modeling [58, 59]. No significant differences between studies emerged, confirming the appropriateness of combining the data. (The HLM results are provided in S2 Analyses.)

## Results

**Mediation.** The association between future focus and feedback effectiveness could be mediated by the effects of attribution disagreement and/or feedback favorability. Specifically, it could be that perceiving the conversation as more future focused is associated with closer agreement on attributions or with perceiving the feedback as more favorable, and one or both of those latter two effects leads to improved feedback effectiveness. Tests of mediation,

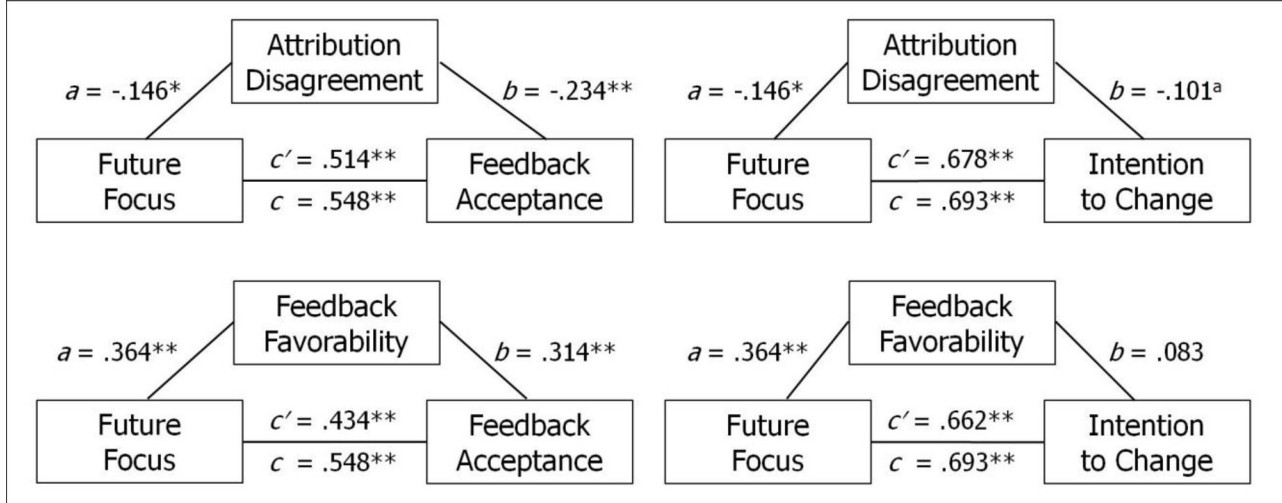

**Fig 4. Direct and indirect effects of perceived future focus on feedback effectiveness.** The two feedback effectiveness measures are feedback acceptance and intention to change. Following Kenny (2018), standardized regression coefficients are shown for the relations between future focus and two hypothesized mediators, attribution disagreement and feedback favorability (*a*), the mediators and the feedback effectiveness measures controlling for future focus (*b*), future focus and the effectiveness measures (*c*), and future focus and the effectiveness measures controlling for the mediator (*c*′). The total effect (*c*) equals the direct effect (*c*′) plus the indirect effect (*a* · *b*). Data are from Studies 2 and 3. [a] *p* = .072; *p* = .028; **p* < .001.

following the methods of Kenny and colleagues [60], suggest otherwise (see Fig 4). These analyses partition the total associations of future focus with feedback acceptance and with intention to change into direct effects and indirect effects. Indirect effects via reduced attribution disagreement were 6.2% of the relation of future focus to feedback acceptance and 2.2% to intention to change. Indirect effects via improved perceptions of feedback favorability were 20.8% of the relation of future focus to feedback acceptance and 4.5% to intention to change. Thus, there is little to suggest that closer agreement on attributions or improved perceptions of feedback favorability account for the benefits of future focus on feedback effectiveness.

**Interactions.** Future focus might have synergistic or moderating effects. In particular, we hypothesized that perceiving the conversation as more future focused may moderate the negative impact of attribution disagreement on feedback effectiveness. Alternatively, future focus may be especially beneficial when agreement about attributions is good, or when attribution differences are neither so big that they cannot be put aside, nor so small that the parties see eye to eye even when they focus on the past. Similarly, future focus may be especially beneficial when feedback is most unfavorable to the recipient, or when it's most favorable, or when it is neither so negative that the recipients can't move past it, nor so positive that the recipients accept it even when the conversation focuses on the past.

We conducted regression analyses with feedback acceptance and intention to change as dependent variables and future focus, feedback favorability, attribution disagreement, and their first-order interactions as predictors. Because some plausible interactions are nonlinear, we defined low, intermediate, and high values for each of the three predictor variables, dividing the 198 participants as evenly as possible for each. We then partitioned each predictor into linear and quadratic components with one degree of freedom each. With linear and quadratic components of three predictors plus a binary variable for Study 2 vs. Study 3, there were seven potential linear effects and 18 possible two-way interactions. We used a stepwise procedure to select which interactions to include in our regressions, using an inclusion parameter of *p* < .15. Results are shown in Table 5.

**Table 5. Stepwise regression results using combined data from Studies 2 and 3.**

|  | Feedback acceptance | | | Intention to change | | |
|---|---|---|---|---|---|---|
|  | *Beta* | *t(188)* | *p* | *Beta* | *t(187)* | *p* |
| Future focus—Linear | 0.487 | 5.09 | < .001 | 0.639 | 11.51 | < .001 |
| Future focus—Quadratic | 0.024 | 0.40 | .687 | -0.068 | -1.27 | .206 |
| Feedback favorability—Linear | 0.268 | 4.36 | < .001 | 0.096 | 1.74 | .083 |
| Feedback favorability—Quadratic | -0.067 | -1.12 | .265 | -0.029 | -0.55 | .584 |
| Attribution disagreement—Linear | -0.226 | -3.57 | .001 | -0.148 | -2.60 | .010 |
| Attribution disagreement—Quadratic | -0.094 | -1.62 | .108 | -0.088 | -1.69 | .093 |
| Study 2 vs. 3 | 0.073 | 1.13 | .259 | -0.078 | -1.34 | .182 |
| Future focus—Linear x Feedback favorability—Linear | -0.119 | -1.91 | .057 | -0.116 | -2.09 | .038 |
| Future focus—Linear x Attribution disagreement—Linear |  |  |  | -0.095 | -1.83 | .070 |
| Future focus—Linear x Study | -0.136 | -1.46 | .145 |  |  |  |
| Feedback favorability–Quadratic x Attribution disagreement–Quadratic |  |  |  | 0.084 | 1.60 | .112 |

Models include all main effects and those first-order interactions that met an entry criterion of *p* < .15, plus data source (Study 2 vs. Study 3). Statistically significant values are underlined.

Future focus interacted with feedback favorability—marginally for feedback acceptance and significantly for intention to change. As shown in Fig 5, recipients who gave low or intermediate ratings for future focus accepted the feedback less when it was most negative ($t(128) = 5.21$, $p < .001$) and similarly, reported less inclination to change ($t(128) = 3.23$, $p = .002$). In contrast, the recipients who rated the feedback discussion as most future focused accepted their

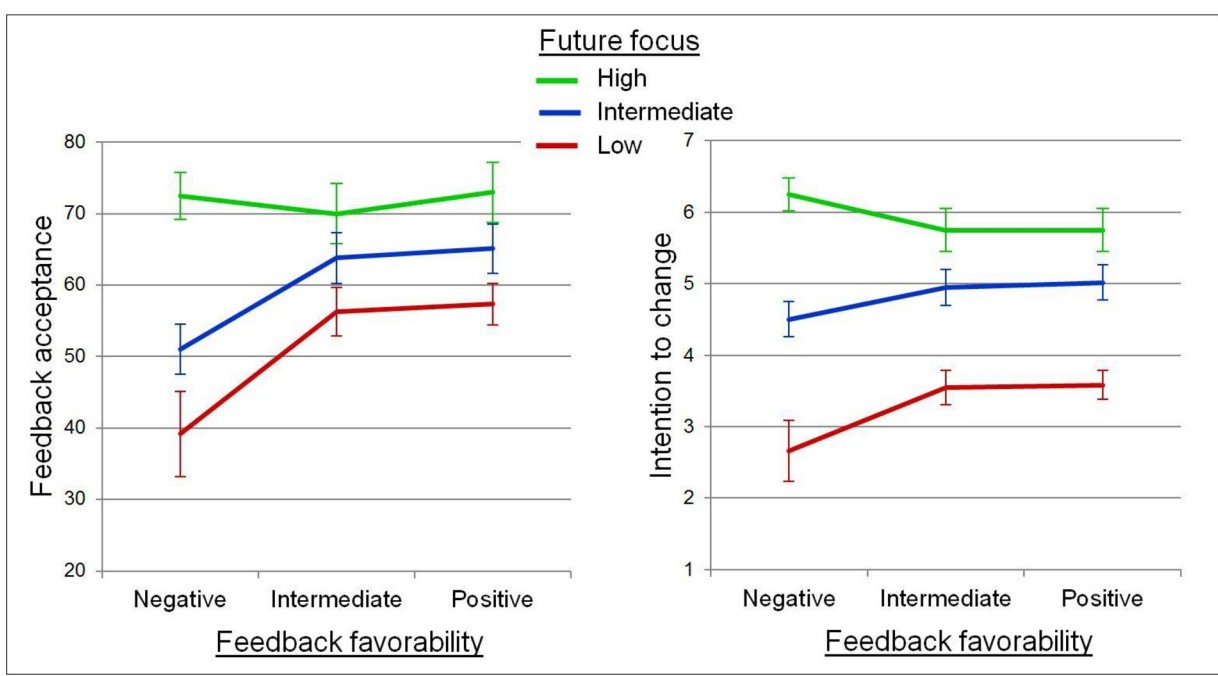

**Fig 5. Recipient ratings of feedback acceptance and intention to change.** Results for each measure of feedback effectiveness are shown by three levels of perceived future focus and three levels of perceived feedback favorability. Error bars show standard errors. Data are from Studies 2 and 3.

feedback and indicated high intention to change at all levels of feedback favorability. These patterns suggest that perceiving future focus moderates the deleterious effect of negative feedback on feedback effectiveness.

On the other hand, we find no evidence that future focus moderates the negative effect of attribution disagreement on feedback effectiveness. Future focus did interact marginally with attribution disagreement for intention to change. However, the benefits of perceiving high vs. low future focus may, in fact, be stronger when there is closer agreement about attributions: The increase in intention to change between low and high future focus groups was 2.30 with high disagreement, 2.37 with intermediate disagreement, and 3.24 in dyads with low disagreement, on a scale from 1 to 7.

**Regression-tree analyses.**   Regression-tree analyses can provide additional insights into the non-linear relations among variables [61], with a better visualization of the best and worst conditions to facilitate feedback acceptance and intention to change. These analyses use the predictors (here, future focus, attribution disagreement, and feedback favorability) to divide participants into subgroups empirically, maximizing the extent to which values on the dependent measure are homogeneous within subgroups and different between them. We generated regression trees for each of our two effectiveness measures, feedback acceptance and intention to change. Fig 6 shows the results, including all subgroups (nodes) with N = 10 or more.

Both trees show that future focus is the most important variable, dividing into lower and higher branches at Nodes 1 and 2, and further distinguishing highest-future groups at Nodes A8 and B6. These representations also reinforce the conclusion that perceived future focus does not operate mainly via an association with more positive feedback or with better agreement on attributions. However, attribution disagreement does play a role, with more agreement leading to better acceptance of feedback and greater intention to change, as long as future focus is at least moderately high (Nodes A3 vs. A4 and B7 vs. B8). (The lack of effect at Node B6 is likely a ceiling effect.) Unfavorable feedback makes matters worse under adverse conditions: when future focus is low (Nodes B3 vs. B4) or when future focus is moderate but attribution disagreement is large (nodes A5 vs. A6).

## General discussion

Our research was motivated by a need to understand why performance feedback conversations do not benefit performance to the extent intended and what might be done to improve that situation. We investigated how providers and recipients of workplace feedback differ in their judgements about the causes of performance and the credibility of feedback, and how feedback discussions impact provider-recipient (dis)agreement and feedback effectiveness. We were particularly interested in how interpretations of past performance, feedback acceptance, and intention to change are affected by the recipient's perception of temporal focus, that is, the extent to which the feedback discussion focuses on past versus future behavior.

Management theorists typically advocate evaluating performance relative to established goals and standards, diagnosing the causes of substandard performance, and providing feedback so that people can learn from the past [19]. They also posit that feedback recipients must recognize there is a problem, accept the feedback as accurate, and find the feedback providers fair and credible in order for performance feedback to motivate improvement [7, 14, 35]. Unfortunately, we know that performance feedback often does not motivate improvement [4]. Our research contributes in several ways to understanding why that is and how feedback conversations might be made more effective.

Decades of attribution theory and research have elucidated the biases thought to produce discrepant explanations for performance between the providers and recipients of feedback.

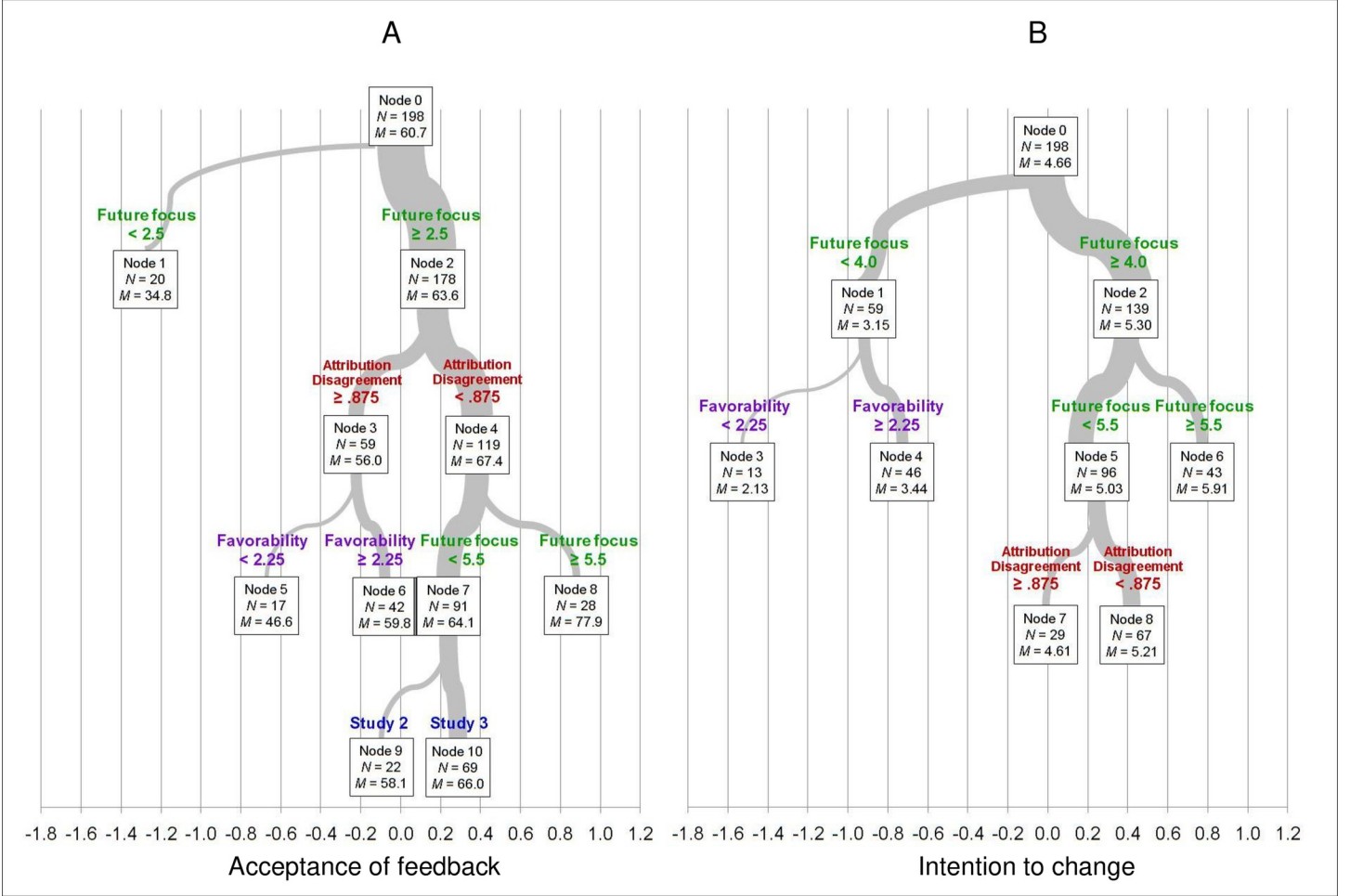

**Fig 6. Regression trees for recipients' ratings of feedback acceptance (0–100) and intention to change (1–7).** The trees depict the effects of future focus, attribution disagreement, and feedback favorability on our two measures of feedback effectiveness. The width of branches is proportional to the number of participants in that branch. Node 0 is the full sample of 198. Values on the X axis are standardized values for each dependent measure. Data are from Studies 2 and 3.

We show that for negative feedback, these discrepancies are prevalent in the workplace. We also show that larger attribution discrepancies are associated with greater rejection of feedback and, in our performance review simulations, with weaker intention to change. These findings support recent research and theory linking performance feedback, work-related decision making, and attribution theory: Instead of changing behavior in response to mixed or negative feedback, people make self-enhancing and self-protecting attributions and judgements they can use to justify not changing [8, 14, 62].

Our research suggests that the common practice of discussing the employees' past performance, with an emphasis on how and why outcomes occurred and what that implies about the employees' strengths and weaknesses, can be counterproductive. Although the parties to a feedback discussion may agree reasonably well about which goals and standards were met or unmet, they are unlikely to converge on an understanding of the causes of unmet goals and standards, even with engaged give and take. Instead, the feedback conversation creates or exacerbates disagreement about the causes of performance outcomes, leading feedback recipients to take more credit for their successes and less responsibility for their failures. This suggests that feedback conversations that attempt to diagnose past performance act as another form of

self-threat that increases the self-serving bias [33]. Surely this runs counter to what the feedback provider intended.

At the same time, we find that self-serving attributions need not stand in the way of feedback acceptance and motivation to improve. A key discovery in our research is that the more recipients feel the feedback focuses on next steps and future actions, the more they accept the feedback and the more they intend to act on it. In fact, when feedback is perceived to be highly future focused, feedback recipients respond as well to predominantly negative feedback as to predominantly positive feedback. Future focus does not nullify self-serving attributions and their detrimental effects [see also 63], but it does enable productive feedback discussions despite them.

We used two complementary research methods. Study 1 used a more naturalistic and thus more ecologically valid method, collecting retrospective self-reports from hundreds of managers about actual feedback interactions in a wide variety of work situations [see 64]. Studies 2 and 3 used a role-play method that allowed us to give all participants identical workplace performance information, a good portion of which was undisputed and quantitative. With that design, response differences between the providers and recipients of feedback are due entirely to role, unconfounded by differences in knowledge and experience.

What role plays cannot establish is the magnitude of effects in organizational settings. Attribution misalignment and resistance to feedback might easily be much stronger in real workplace performance reviews where it would be rare for the parties to arrive with identical, largely unambiguous information. Moreover, managers' investment in the monetary and career outcomes of performance reviews might lead feedback recipients to feel more threatened than in a role play and thus to disagree even more with unfavorable feedback. On the other hand, the desire to maintain employment and/or to maintain good relationships with supervisors might motivate managers to re-assess their past achievements, to change their private attributions, and to be more accepting of unfavorable feedback. Data from our role-play studies may not speak to the magnitude of resistance to feedback in work settings (although our survey results suggest it's substantial), but they do show that feedback acceptance is increased when the participants perceive their feedback to be focused on the future.

## Implications for future research and theory

There are few research topics more important to the study of organizations than performance management. Feedback conversations are a cornerstone of most individual and team performance management, yet there is still much we do not know about what should be said, how, and why. Based on research into the motivational advantages of prospective thinking, we hypothesized that feedback discussions perceived as future focused are the most effective kind for generating acceptance of feedback and fostering positive behavior change. Our findings support that hypothesis. The present research contributes to the literature on prospection by highlighting the role of interpersonal interactions in facilitating prefactual thinking and any associated advantages for goal pursuit [39, 43–45, 63, 65]. In this section we suggest three lines of future research: (a) field studies and interventions; (b) research into the potential role of self-beliefs; and (c) exploration of the conversational dynamics associated with feedback perceived as past vs. future focused.

**Field research and intervention designs.** Testing feedback interventions in the workplace and other field settings is an important future step toward corroborating, elaborating, or correcting our findings. It will be necessary to develop effective means to foster a more future-focused style of feedback. Then, randomized controlled trials that contrast future-focused with diagnostic feedback can demonstrate the benefits that may accrue from focusing feedback

more on future behavior and less on past behavior. Participant evaluations of the feedback discussions can be supplemented by those of neutral observers. Such evaluations are directly relevant to organizational goals, including employee motivation, positive supervisor-supervisee relations, and effective problem solving. Assessing subsequent behavior change and job performance is both important and complicated for evaluating feedback effectiveness: Seeing intentions through to fruition depends on many factors, including individual differences in self-regulation [66, 67] and factors beyond people's control, such as competing commitments, limited resources, and changing priorities [68–71]. Nevertheless, the ultimate proof of future-focused feedback will lie in performance improvement itself.

**Self-beliefs and future focus.**   If future focus enhances feedback effectiveness, it may do so via self-beliefs. Growth mindset and self-efficacy, for example, are self-beliefs that influence how people think about and act on the future. Discussions that focus on what people can do in the future to improve performance may encourage people to view their own behavior as malleable and to view better results as achievable. If future focus helps people access this growth mindset, it should orient them toward mastering challenges and improving the self for the future: Whereas people exercise defensive self-esteem repair when in a fixed mindset, they prefer self-improvement when accessing a growth mindset [72, 73]. Similarly, feedback conversations that focus on ways the feedback recipient can attain goals in the future may enhance people's confidence in their ability to execute the appropriate strategies and necessary behaviors to succeed. Such self-efficacy expectancies have been shown to influence the goals people select, the effort and resources they devote, their persistence in the face of obstacles, and the motivation to get started [74, 75]. Thus, research is needed to assess whether future focus alters people's self-beliefs (or vice versa; see below) and if these, in turn, impact people's acceptance of feedback and intention to change.

We found sizeable variation in the extent to which dyads reported focusing on the future. Pre-existing individual differences in self-beliefs may contribute to that variation. Recent research, for example, finds that professors with more growth mindsets have students who perform better and report being more motivated to do their best work [76]. In the case of a feedback conversation, we suspect that either party can initiate thinking prospectively, but both must participate in it to sustain the benefits.

**Conversational dynamics and future focus.**   Unlike most studies of people's reactions to mixed or negative feedback, our studies use face-to-face, real-time interaction, that is to say, two people in conversation. Might conversational dynamics associated with future-focused feedback contribute to its being better accepted and more motivating than feedback focused on the past? Do managers who focus more on the future listen to other people's ideas and perspectives in ways that are perceived as more empathic and nonjudgmental? Do these more prospective discussions elicit greater cooperative problem solving? Research on conversation in the workplace is in its early stages [77], but some studies support the idea that high quality listening and partner responsiveness might reduce defensiveness, increase self-awareness, or produce greater willingness to consider new perspectives and ideas [78, 79].

## Practical implications

Our studies provide the first empirical evidence that managers can make feedback more effective by focusing it on the future. Future-focused feedback, as we define it, is characterized by prospective thinking and by collaboration in generating ideas, planning, and problem-solving. We assessed the degree of future focus by asking participants to rate the extent to which the feedback discussion focused on future behavior, the two parties spent time generating new ideas for next steps, and the conversation centered on how to make the recipient successful. This differs greatly from feedback research that distinguishes past vs. future orientation "using

minimal rewording of each critique comment" (e.g., you didn't always demonstrate awareness of. . . vs. you should aim to demonstrate more awareness of. . .) [80 p. 1866].

Because future-focused feedback is feedback, it also differs from both advice giving and "feedforward" (although it might be advantageous to incorporate these): It differs from Kluger and Nir's feedforward interview, which queries how the conditions that enabled a person's positive work experiences might be replicated in the future [81], and from Goldsmith's feedforward exercise, which involves requesting and receiving suggestions for the future, without discussion or feedback [82].

The scenario at the very start of this article asks, "What can Chris say to get through to Taylor?" A future-focused answer might include the following: Chris first clarifies that the purpose of the feedback is to improve Taylor's future performance, with the goal of furthering Taylor's career. Chris applauds Taylor's successes and is forthright and specific about Taylor's shortcomings, while avoiding discussion of causes and explanations. Chris signals belief that Taylor has the motivation and competence to improve [83]. Chris then initiates a discussion in which they work together to develop ideas for how Taylor can achieve better outcomes in the future. (For a more detailed illustration of a future-focused conversation, see S11 Text.)

## Conclusions

Our research supports the intriguing possibility that the future of feedback could be more effective and less aversive than its past. Performance management need not be tied to unearthing the determinants of past performance and holding people to account for past failures. Rather, performance may be managed most successfully by collaborating with the feedback recipient to generate next steps, to develop opportunities for interesting and worthwhile endeavors, and to enlarge the vision of what the recipient could accomplish. Most organizations and most managers want their workers to perform well. Most workers wish to succeed at their jobs. Everyone benefits when feedback discussions develop new ideas and solutions and when the recipients of feedback are motivated to make changes based on those. A future-focused approach to feedback holds great promise for motivating future performance improvement.

## Supporting information

**S1 Text. Study 1 instructions and survey.**
(PDF)

**S2 Text. Study 2 instructions.**
(DOCX)

**S3 Text. Study 2 pre-discussion questionnaire–Regional Manager.**
(DOCX)

**S4 Text. Study 2 pre-discussion questionnaire–District Manager.**
(DOCX)

**S5 Text. Study 2 post-discussion questionnaire–Regional Manager, pre-post group.**
(DOCX)

**S6 Text. Study 2 post-discussion questionnaire–District Manager, pre-post group.**
(DOCX)

**S7 Text. Study 2 post-discussion questionnaire–both roles, post-only group.**
(DOCX)

**S8 Text. Study 3 guidelines added to instructions–two versions.**
(DOCX)

**S9 Text. Study 3 post-discussion questionnaire–Regional Manager.**
(DOCX)

**S10 Text. Study 3 post-discussion questionnaire–District Manager.**
(DOCX)

**S11 Text. Hypothetical example of future-focused feedback.**
(DOCX)

**S1 Table. Study 1 correlations among variables.**
(XLSX)

**S2 Table. Study 2 means and standard deviations for evaluations of successes and failures, by role.**
(DOCX)

**S3 Table. Studies 2 and 3 correlations among variables.**
(XLSX)

**S1 Analyses. Study 2 ANOVAs with between-groups comparisons.**
(PDF)

**S2 Analyses. Studies 2 and 3 HLM analyses.**
(DOCX)

**S1 Dataset. Studies 1, 2 and 3 data.**
(XLSX)

## Acknowledgments

For helpful comments on earlier drafts of this paper, we are grateful to Pino Audia, Angelo Denisi, Nick Epley, Ayelet Fishbach, Brian Gibbs, Reid Hastie, Chris Hsee, Remus Ilies, David Nussbaum, Jay Russo, Paul Schoemaker, William Swann, and Kathleen Vohs.

## Author Contributions

**Conceptualization:** Jackie Gnepp, Joshua Klayman, Ian O. Williamson.

**Data curation:** Joshua Klayman, Ian O. Williamson.

**Formal analysis:** Jackie Gnepp, Joshua Klayman, Ian O. Williamson, Sema Barlas.

**Funding acquisition:** Ian O. Williamson.

**Investigation:** Jackie Gnepp, Joshua Klayman, Ian O. Williamson.

**Methodology:** Jackie Gnepp, Joshua Klayman, Ian O. Williamson.

**Project administration:** Jackie Gnepp, Joshua Klayman, Ian O. Williamson.

**Resources:** Jackie Gnepp, Joshua Klayman, Ian O. Williamson.

**Software:** Joshua Klayman, Sema Barlas.

**Supervision:** Jackie Gnepp, Ian O. Williamson.

**Validation:** Jackie Gnepp, Joshua Klayman, Ian O. Williamson.

**Visualization:** Jackie Gnepp, Joshua Klayman, Sema Barlas.

**Writing – original draft:** Jackie Gnepp, Joshua Klayman.

**Writing – review & editing:** Jackie Gnepp, Joshua Klayman, Ian O. Williamson, Sema Barlas.

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
