## [Decision Letter · Decision Letter 0]

7 Apr 2020

PONE-D-20-05644

The future of feedback:  Motivating performance improvement

PLOS ONE

Dear Dr Klayman,

Thank you for submitting your manuscript to PLOS ONE. After careful consideration, we feel that it has merit but does not fully meet PLOS ONE’s publication criteria as it currently stands. Therefore, we invite you to submit a revised version of the manuscript that addresses the points raised during the review process.

We would appreciate receiving your revised manuscript by May 22 2020 11:59PM. To enhance the reproducibility of your results, we recommend that if applicable you deposit your laboratory protocols in protocols.io, where a protocol can be assigned its own identifier (DOI) such that it can be cited independently in the future. For instructions see: http://journals.plos.org/plosone/s/submission-guidelines#loc-laboratory-protocols

We look forward to receiving your revised manuscript.

Kind regards,

Paola Iannello

Academic Editor

PLOS ONE

2. Please modify the title to ensure that it is meeting PLOS’ guidelines (https://journals.plos.org/plosone/s/submission-guidelines#loc-title). In particular, the title should be "specific, descriptive, concise, and comprehensible to readers outside the field" and in this case it is not informative and specific about your study's scope and methodology.

We note that one or more of the authors are employed by a commercial company: Humanly Possible, Inc.

Reviewers' comments:

Reviewer's Responses to Questions

**Comments to the Author**

1. Is the manuscript technically sound, and do the data support the conclusions?

Reviewer #1: Yes

Reviewer #2: Yes

2. Has the statistical analysis been performed appropriately and rigorously? 

Reviewer #1: Yes

Reviewer #2: Yes

3. Have the authors made all data underlying the findings in their manuscript fully available?

Reviewer #1: Yes

Reviewer #2: Yes

4. Is the manuscript presented in an intelligible fashion and written in standard English?

Reviewer #1: Yes

Reviewer #2: Yes

5. Review Comments to the Author

Reviewer #1: 1. I enjoyed reading this manuscript, but it appears to be unnecessary long in parts and readability would benefit of a more concise style. I would recommend condensing some parts, for example in the methods section for study 2 was overly long and lacked clarity in parts. The description of the second questionnaire was a little confusing in terms of the consistency in how items were measured and the hypothesis was not clear.

2. In the ethics statement for Study 1 (line 184), please explain the rationale behind the waiver of consent.

3. Procedure (line 187) please give details of the survey platform used.

4. Results -Please include the number of participants in each group.

5. Please comment on what normality checks were performed to assess the distribution of the data.

6. Line 470, correlations are discussed but I can’t see a table to support these.

7. The discussion did not address the results in relation to previous literature and lacked a theoretical explanation of the findings (See for example ‘Korn CW, Rosenblau G, Rodriguez Buritica JM, Heekeren HR (2016) Performance Feedback Processing Is Positively Biased As Predicted by Attribution Theory. PLoS ONE 11(2)’ for a discussion of attributional style and self-serving bias. I recommend some rewrite of the discussion with more reference to theory.

8. Some acknowledgement of the effect of individual differences in self-regulation would be useful to include as this may influence how feedback is received in terms of attributions. See for example, ‘Donovan, JJ, Lorenzet, SJ, Dwight, SA, Schneider, D. The impact of goal progress and individual differences on self‐regulation in training. J Appl Soc Psychol. 2018; 48: 661– 674’.

9. The suggestions for improvement at the end of the study would be better to be condensed to give a brief suggestion of methods.

Reviewer #2: The paper reports an interesting and comprehensive work about a relevant issue in organizational psychology. Both the theoretical frame and the applied methodology are original and thorough, though the use of role-play raises some doubts about the robustness of the results (some concerns are raised by the authors themselves (lines 752-760) ). This is, in my opinion, the main limitation of studies 2 and 3. I would suggest that the authors insert a wider reasoning about the choice of using this method to collect their data and the pros and cons.

In the "General Discussion" paragraph the authors state that "We investigated the sources of agreement and disagreement between feedback provider and recipient" (lines 712-713). I strongly suggest that this sentence is being modified, since it doesn't describe the aim nor the results in Study 1 correctly.

6. PLOS authors have the option to publish the peer review history of their article (what does this mean?). If published, this will include your full peer review and any attached files.

Reviewer #1: No

Reviewer #2: Yes: Federica Biassoni

---

## [Author Response · Author response to Decision Letter 0]

12 May 2020

Please see uploaded document Response to Reviewers. Text copied here.

Response to Reviewers

PONE-D-20-05644

The future of feedback: Motivating performance improvement through future-focused feedback

PLOS ONE

We wish to thank the reviewers for their very helpful and constructive comments. We especially appreciate the clarity and specificity with which they framed their suggestions. Below we respond to each reviewer recommendation.

Reviewer #1:

1. I enjoyed reading this manuscript, but it appears to be unnecessary long in parts and readability would benefit of a more concise style. I would recommend condensing some parts, for example in the methods section for study 2 was overly long and lacked clarity in parts. The description of the second questionnaire was a little confusing in terms of the consistency in how items were measured and the hypothesis was not clear.

We revised the methods section for Study 2 (former lines 274-279; 285-414, revision lines 276-281; 299-402). The new version is a full page shorter and, in line with the reviewer’s suggestion, we believe this more concise version is now more readable. It includes a revised description of the post-discussion questionnaires (former 346-367; revision 350-361), clarifying the sequence and types of questions provided to each group. It also includes revisions, mainly in the Design section (former 387-414; revision lines 377-402) to clarify how the various measures related to our hypotheses. 

2. In the ethics statement for Study 1 (line 184), please explain the rationale behind the waiver of consent.

Study 1 was approved by the Institutional Review Board at the University of Chicago, which waived the requirement for written consent as was its customary policy for studies judged to be minimal risk, involving only individual, anonymized survey responses. Their decision cited US Code 45 CFR 46.101(b). Citing the code in our manuscript seemed overly legalistic, but we have added the rest of the rationale to the ethics statement (former lines 184-185; revision 184-186).

3. Procedure (line 187) please give details of the survey platform used.

We now identify the platform as Cogix ViewsFlash (revision line 188).

4. Results -Please include the number of participants in each group.

 We have added the requested information for Study 1 (revision lines 214-215). Following up on the suggestion, we also made it easier to locate the corresponding information for Study 2 (revision lines 316-317).

5. Please comment on what normality checks were performed to assess the distribution of the data.

The general consensus is that the analyses we use, i.e. ANOVA and linear regression, are generally quite robust with regard to moderate violations of normality with Ns on the order of ours (e.g., Blanca, Alarcón, Arnau, Bono, & Bendayan, Psichothema, 2017; Schmidt & Finana, Journal of Clinical Epidemiology, 2018; Ali & Sharma, Journal of Econometrics, 1996; Schmider, Ziegler Danay, Beyer, & Bühner, Methodology, 2010). Nevertheless, we used an arcsine transformation on the variables a priori most likely to suffer from systematic deviations, namely the attribution proportions. Most authors recommend checking for major deviations from normality by plotting model-predicted values against residuals and against the normal distribution (using P-P or Q-Q plots). We did that for our analyses (graphs attached), and found no troublesome deviations, with the possible exception of one variable of minor importance to our main results or theory, namely performance quality ratings for successes in Study 2. We note in the paper that that variable may suffer from ceiling effects (former 468-469, revision 456-457). We did not add a discussion of normality to the paper because of the increased length and complexity that would involve and because it’s seldom an issue of concern with data and analyses like ours. However, we could include the graphs we’ve attached here as supplemental material if you tell us you would like us to do so. 

6. Line 470, correlations are discussed but I can’t see a table to support these.

Thank you for alerting us to this inadvertent omission. We now include complete correlation tables for all the variables analyzed in each Study in the supplemental materials: S2 Table for Study 1 (revision lines 224-225) and S11 Tables for Studies 2 and 3 separately and combined (revision lines 458-459), with provider-recipient correlations identified by color shading. (S2 was formerly the dataset for Study 1, but now data from all three studies are contained in S17.)

7. The discussion did not address the results in relation to previous literature and lacked a theoretical explanation of the findings (See for example ‘Korn CW, Rosenblau G, Rodriguez Buritica JM, Heekeren HR (2016) Performance Feedback Processing Is Positively Biased As Predicted by Attribution Theory. PLoS ONE 11(2)’ for a discussion of attributional style and self-serving bias. I recommend some rewrite of the discussion with more reference to theory.

To better address our results in relation to previous attribution literature and theory, we have revised former lines 723-740 in the General Discussion. Now we more clearly discuss our findings in relation to self-serving bias, self-threat, and both historical and more recent formulations of attribution theory, including the helpful reference the reviewer provided (revision lines 708-735). We have also added a brief discussion of how our results relate to previous literature on future thinking (revision lines 760-762). We attempted to minimize redundancy with the Introduction section. The new material includes several new references.

8. Some acknowledgement of the effect of individual differences in self-regulation would be useful to include as this may influence how feedback is received in terms of attributions. See for example, ‘Donovan, JJ, Lorenzet, SJ, Dwight, SA, Schneider, D. The impact of goal progress and individual differences on self‐regulation in training. J Appl Soc Psychol. 2018; 48: 661– 674’.

We added mention in the General Discussion of individual differences in self-regulation, citing two references, including the one helpfully provided by Reviewer #1 (revision line 776). Additionally, we reworded former lines 798-799 (revision lines 793-794) to make it clearer that we are acknowledging individual differences there as well.

9. The suggestions for improvement at the end of the study would be better to be condensed to give a brief suggestion of methods.

We condensed former lines 828-846 from 19 lines to 8 lines (revision lines 823-830), referring the interested reader to new Supporting Information S16 Text for the expanded version. We trust this solution meets the recommendation for a brief suggestion of methods, while also satisfying the interests of those seeking more detail. 

Reviewer #2: 

1. The paper reports an interesting and comprehensive work about a relevant issue in organizational psychology. Both the theoretical frame and the applied methodology are original and thorough, though the use of role-play raises some doubts about the robustness of the results (some concerns are raised by the authors themselves (lines 752-760)). This is, in my opinion, the main limitation of studies 2 and 3. I would suggest that the authors insert a wider reasoning about the choice of using this method to collect their data and the pros and cons.

We now include a wider reasoning about our choice to use a role-play method and the pros and cons. The new version comprises revision lines 282-298. (We also revised the subsequent paragraph for increased clarity, given the insertion of the new paragraph about the role-play method.)

2. In the "General Discussion" paragraph the authors state that "We investigated the sources of agreement and disagreement between feedback provider and recipient" (lines 712-713). I strongly suggest that this sentence is being modified, since it doesn't describe the aim nor the results in Study 1 correctly.

Thank you for your careful reading. We have re-written that sentence to more accurately capture the results of Study 1 as well as the other two studies (revised lines 697-700).

[Figures attached--please see uploaded document Response to Reviewers.]

---

## [Decision Letter · Decision Letter 1]

27 May 2020

The future of feedback: Survey and role-play investigations into causal attributions, feedback acceptance, motivation to improve, and the potential benefits of future focus for increasing feedback effectiveness in the workplace

PONE-D-20-05644R1

Dear Dr. Klayman,

We are pleased to inform you that your manuscript has been judged scientifically suitable for publication and will be formally accepted for publication once it complies with all outstanding technical requirements.

With kind regards,

Paola Iannello

Academic Editor

PLOS ONE

Additional Editor Comments (optional):

Reviewers' comments:

Reviewer's Responses to Questions

**Comments to the Author**

1. If the authors have adequately addressed your comments raised in a previous round of review and you feel that this manuscript is now acceptable for publication, you may indicate that here to bypass the “Comments to the Author” section, enter your conflict of interest statement in the “Confidential to Editor” section, and submit your "Accept" recommendation.

Reviewer #1: All comments have been addressed

Reviewer #2: All comments have been addressed

2. Is the manuscript technically sound, and do the data support the conclusions?

Reviewer #1: Yes

Reviewer #2: Yes

3. Has the statistical analysis been performed appropriately and rigorously? 

Reviewer #1: Yes

Reviewer #2: Yes

4. Have the authors made all data underlying the findings in their manuscript fully available?

Reviewer #1: Yes

Reviewer #2: Yes

5. Is the manuscript presented in an intelligible fashion and written in standard English?

Reviewer #1: Yes

Reviewer #2: Yes

6. Review Comments to the Author

Reviewer #1: (No Response)

Reviewer #2: (No Response)

7. PLOS authors have the option to publish the peer review history of their article (what does this mean?). If published, this will include your full peer review and any attached files.

Reviewer #1: No

Reviewer #2: Yes: Federica Biassoni

---

## [Editor Report · Acceptance letter]

5 Jun 2020

PONE-D-20-05644R1 

The future of feedback:  Motivating performance improvement through future-focused feedback  

Dear Dr. Klayman:

I'm pleased to inform you that your manuscript has been deemed suitable for publication in PLOS ONE. Congratulations! Your manuscript is now with our production department. 

Kind regards, 

on behalf of

Dr. Paola Iannello 

Academic Editor

PLOS ONE